# A Digital Tool for Measuring Healing of Chronic Wounds Treated with an Antioxidant Dressing: A Case Series

**DOI:** 10.3390/ijerph20054147

**Published:** 2023-02-25

**Authors:** Inés María Comino-Sanz, Rafael Cabello Jaime, Josefina Arboledas Bellón, Juan Francisco Jiménez-García, Mercedes Muñoz-Conde, María José Díez Requena, Francisco Javier García Díaz, Begoña Castro, Pedro Luis Pancorbo-Hidalgo

**Affiliations:** 1Department of Nursing, Faculty of Health Sciences, University of Jaén, 23071 Jaén, Spain; 2Malaga-Axarquia Health District, Andalusian Health Service, Av. del Sol, 43, 29740 Torre del Mar, Málaga, Spain; 3Northeast-Jaen Health District, Andalusian Health Service, Ctra. de Linares Km.1, 23400 Úbeda, Jaén, Spain; 4Poniente-Almeria Health District, Andalusian Health Service, C. Jesus de Perceval, 22, 04700 El Ejido, Almería, Spain; 5Jaen Health District, Andalusian Health Service, Arquitecto Berges N°10, 23006 Jaén, Spain; 6Histocell S.L., Bizkaia Science and Technology Park, 48160 Derio, Bizkaia, Spain

**Keywords:** wound healing, oxidative stress, antioxidant dressing, digital tool

## Abstract

(1) Abstract: Wound monitoring is an essential aspect in the evaluation of wound healing. This can be carried out with the multidimensional tool HELCOS, which develops a quantitative analysis and graphic representation of wound healing evolution via imaging. It compares the area and tissues present in the wound bed. This instrument is used for chronic wounds in which the healing process is altered. This article describes the potential use of this tool to improve the monitoring and follow-up of wounds and presents a case series of various chronic wounds with diverse etiology treated with an antioxidant dressing. (2) Methods: A secondary analysis of data from a case series of wounds treated with an antioxidant dressing and monitored with the HELCOS tool. (3) Results: The HELCOS tool is useful for measuring changes in the wound area and identifying wound bed tissues. In the six cases described in this article, the tool was able to monitor the healing of the wounds treated with the antioxidant dressing. (4) Conclusions: the monitoring of wound healing with this multidimensional HELCOS tool offers new possibilities to facilitate treatment decisions by healthcare professionals.

## 1. Introduction

A chronic wound, also called a hard-to-heal wound, has been defined as any wound that has not healed by 40–50% after four weeks of appropriate treatment [1]. Several factors can delay the physiological process of healing, including oxygenation, infection, diabetes, medications, stress, nutrition, hormones, and age [2].

When assessing the wound healing process, clinicians often face the problem of reliably measuring wound size. Wound measurement is important for monitoring the healing process of chronic wounds and to evaluate the effect of treatments. This is a practical problem as most of the measures used are subjective and based on the clinical experience of the professional.

In the last few decades, technological advances have led to the development of several accurate methods and multidimensional tools for wound monitoring: manual or digital planimetry, simple ruler method, mathematical models, digital imaging, or more recently three-dimensional (3D) [3]. As a result, wound monitoring is more objective and allows the identification of different parameters and variables through a specific analysis.

One of the multidimensional assessment tools recently developed is the HELCOS software, a web-based integrated wound management system that allows the measurement of different wound parameters through digital analysis of images of the wounds [4]. HELCOS was designed and developed between 2015 and 2017 through a project funded by the Spanish Pressure Ulcer and Chronic Wound Advisory Group. This tool has been available free of charge since 2017 for clinicians working in clinical settings; only a short registration is required. There are no special hardware requirements to use this tool, only a computer or device connected to the Internet, so it can be used directly in any environment (hospital, wound clinic, primary care). All personal data security standards are guaranteed; each professional can only access his/her own cases. To perform a wound analysis, the clinician has to obtain an image of the wound with any type of camera or device. Good lighting conditions are highly recommended, taking the picture at 20 to 30 cm, perpendicular to the wound plane and placing a size test of known diameter close to the wound (such as a blue circle 2 cm in diameter). Photos can be uploaded directly from a smartphone or using a computer.

HELCOS allows clinicians to measure the wound area and the proportion of the wound bed covered with granulation or necrotic tissue. We have tested this tool in a series of wound cases treated with an antioxidant dressing.

It is known that wound healing is impaired when the wound remains in the inflammatory stage for too long [5]. Oxidative stress is among the factors that can delay the healing process [2]. Reactive oxygen species (ROS) are small oxygen-derived molecules that play a crucial role in the preparation of the normal wound healing response [6]. Therefore, a suitable balance between the levels of ROS is essential. A wound with a low level of ROS protects tissues against infection and stimulates effective wound healing by promoting cell survival [7,8], whereas if there is excess ROS in the wound, the cells are damaged with pro-inflammatory status and produce oxidative stress [9].

Therefore, the use of antioxidant compounds for wound treatment is increasing and has excellent potential for clinical use. Antioxidant dressings that regulate this balance are a target for new therapies [10,11]. Among these new advanced products is the antioxidant dressing Reoxcare^®^ [12], developed by Histocell (Bizkaia, Spain). This product combines an absorbent matrix obtained from the locust bean gum galactomannan, of plant-based origin, with an antioxidant hydration solution with curcumin and N Acetylcysteine (NAC) [13].

Curcumin is a natural phenol extracted from the Curcuma longa rhizome. It has anti-inflammatory, antibacterial, and antioxidant properties, which improve wound healing [14]. NAC is an antioxidant molecule that plays an important role in regulating redox status [15]. The three components act synergistically, giving the product a potent antioxidant activity. Due to the innovative design, this antioxidant dressing combines the advantages of moist healing in exudate management and free radical neutralization, achieving wound reactivation.

This antioxidant dressing was tested in different studies. In vitro studies and animal wound healing models have shown that this product modulates the inflammatory phase of wound healing, controlling the excessive cell activation and allowing a more orderly transition between the inflammatory, proliferative, and remodeling phases of wound healing [13]. A multicenter, prospective-case study series revealed that this dressing can be applied to wounds independently of their level of recurrence or severity, effectively eliminating the biofilm and facilitating the progression of the wound out of the inflammatory phase [16,17]. These findings suggest that the dressing could be a new advanced alternative for managing hard-to heal wounds. In other words, the value of antioxidant dressing in the management has been reported and shown positive results.

The purpose of this article is to describe the potential use of a web-based wound measurement system (HELCOS) for monitoring the progress of wound healing in a case series of wounds.

## 2. Materials and Methods

### 2.1. Study Design

This consists of a secondary analysis of a case series from the intervention group of the main study. This is a descriptive design of healing monitoring using the HELCOS tool.

The main study is a prospective intervention study with two arms, intervention (antioxidant dressing) and comparison (usual care with moist dressing) [18]. Advanced practice wound nurses recruited patients with chronic wounds in three primary health care centers in the Andalusian Health Service in Spain between September 2019 and October 2021.

The main study included 54 patients (28 intervention group and 26 comparison group). Patients were eligible if they were aged 18 years or older with the following: leg ulcer (venous, ischemic, traumatic, or diabetic foot ulcer), dehisced surgical wound healing by second intention, or pressure ulcers. Wound area was between 1 and 250 cm^2^. Exclusion criteria were systemic inflammatory disease or oncological disease, wounds with clinical signs of infection, terminal situation (life expectancy less than 6 months), pregnancy or wounds treated with negative pressure therapy.

A cut-off of 8 weeks (or healing if this occurred before 8 weeks) was established. A clinical nurse assessed patients at baseline and at weeks 2, 4, 6, and 8 to determine their evolution. Data collected from each patient included demographic characteristics, patient’s clinical background (concomitant medical diagnosis, clinical antecedents, nutritional status, smoking habit), description of the wound (etiology, size, location, specific characteristics), healing measured by RESVECH 2.0 score and variation in wound are measured by HELCOS tool.

#### 2.1.1. Wound Management

Patients were managed according to a good standard of care. A general protocol for wound management was established: cleaning the wound with sterile physiological saline solution, debridement to deep clean the nonviable tissues in the wound bed, antioxidant dressing application as a primary dressing, and cover with secondary dressing. The dressing is kept in place for 2 to 3 days, according to the manufacturer’s recommendations and depending on the level of wound exudates.

#### 2.1.2. Statistical Analysis

Descriptive statistics were used (mean and standard deviation for quantitative variables; frequency and percentages for nominal variables).

#### 2.1.3. Ethical Aspects

The study was approved by the Ethics Committee of Jaen (Andalusian Health System) with reference number 0645-N-19. The study was conducted in accordance with the ethical principles of the Declaration of Helsinki. The patients provided written informed consent, which ensured data confidentiality.

## 3. Results

### 3.1. Description of HELCOS Wound Healing Software

HELCOS is an integrated wound management system that calculates wound area and the relative percentage of tissue types in the wound bed using an image of the lesion. This image is loaded into the system and assigned to a patient and a case. For each patient, different images of the lesion can be attached over time to evaluate its evolution using different methods. This version is free and accessible in Spanish [4].

First, wound area is checked by measuring length and width directly with a graduated ruler (Kundin method) [19]. Then, it is estimated using digital analysis of wound photography.

Second, the relative percentage of tissue types in the wound bed (granulation, slough and necrotic tissue) is estimated. This software identifies tissue types by using different colors in the wound bed: red for granulation tissue, yellow for slough, and black for necrotic tissue. It also creates a graph showing the evolution of the percentage of tissue present in the wound bed over the follow-up period.

In addition, the RESVECH 2.0 scale is integrated in this software for evaluation of the status of the wound [20]. It assesses six aspects (wound size, depth/affected tissues, wound edges, type of tissue in the wound bed, exudate, and infection/inflammation). The score of this scale ranges from 0 points (wound healed) to 35 points (the worst possible status of the wound). A lower score means an improvement in the healing process. This scale is an excellent tool for comparing the data grouped according to the type of wound, recurrence, or severity.

### 3.2. Description of Wounds

In reference to the etiology of the wounds, 28.6% were venous, 7.1% ischemic, 7.1% diabetic, 25% traumatic, 10.7% surgical wound, and 21.4% pressure injuries. The wound locations were 42.9% leg, 39.3% food, 10.7% gluteus/coccyx, 3.6% abdomen, and 3.6% upper limb. The general wound characteristics are presented below (Table 1). Twelve wounds treated with the antioxidant dressing were healed at 8 weeks (42.86%) and 16 had an increase of 50% or more in granulation tissue (57.14%).

### 3.3. Healing Monitoring

We present several significant cases of wounds treated with the antioxidant dressing over eight weeks, which were monitored with the HELCOS software and achieved complete wound healing, significantly reduced wound area, or showed an important change in the tissues present in the wound bed.

The data and graphs presented in each of the cases refer to the analysis of the percentage of tissues present in the wound bed (granulation tissue and devitalized tissue—sloughed or necrotic) and the area of the lesion as analyzed with the HELCOS system and demonstrates wound follow-up.

Case 1. Traumatic leg wound.

A 59-year-old male presented with a traumatic wound on the lower limb, which was not healing (Figure 1). The initial area of the wound was 5.86 cm^2^, with a depth affecting muscle, defined borders, tissue compatible with biofilm, and desquamation on the perilesional skin. At the week 6 assessment, we observed complete wound healing (Table 2).

The tissues present in the wound bed showed a favorable evolution toward healing throughout the 6 weeks of treatment, decreasing the percentage of sloughed tissue present in the wound bed and increasing granulation tissue (Figure 2).

Case 2. Incised leg wound.

A 71-year-old male presented with a traumatic injury to the internal tibial area. This wound had damaged edges and abundant exudate (Figure 3). The initial area was 4.73 cm^2^, with 89.98% devitalized tissue (necrotic/sloughed), and only 10.02% was granulation tissue. Over 8 weeks, the wound area was reduced and the wound bed was cleaned, until reaching complete healing (Table 3) (Figure 4).

Case 3. Wound with venous etiology.

This was a 72-year-old woman with a venous wound in the anterior tibial area. In the initial assessment, the wound area was 12.31 cm^2^, the edges were damaged, and there was a saturation of exudate (Figure 5). The percentage of tissues in the bed was 60.58% granulation tissue and 39.43% sloughed tissue. At 8 weeks, the antioxidant treatment achieved wound closure, contributed to the removal of sloughed tissues, and induced granulation tissue formation (Figure 6). It should be noted that this treatment also significantly reduced pain; at the initial assessment, the patient presented 10/10 on the Visual Analog Scale (VAS), 4/10 at week 2, 1/10 at weeks 4 and 6, and no pain by week 8.

Case 4. Traumatic cavity wound.

This was a 67-year-old male with a cavity wound of traumatic etiology located in the lower extremity. This clinical case stands out for its rapid evolution. The initial area was 5.42 cm^2^, highlighting the depth of the cavitation, but in just four weeks he achieved complete healing and a favorable evolution of the tissues (Figure 7 and Figure 8). It should also be emphasized that initially he reported a 10/10 on the VAS pain scale, which decreased to 5/10 in week 2, and completely disappeared in week 4.

Case 5. Diabetic foot ulcer.

This was a 57-year-old man with a diabetic foot ulcer that had an initial area of 1.54 cm^2^, and closed at week 8 (Figure 9). The antioxidant treatment was able to clean the wound bed, completely eliminating the sloughed tissue and facilitating the production of granulation tissue. At baseline, the wound had 76.19% granulation tissue and 23.81% sloughed tissue; from week 2 to week 8 only granulation tissue was observed in the wound (100%) (Figure 10).

Case 6. Dehiscence surgical wound.

This was a 75-year-old male presented with a surgical wound in the lower limb that was healing by second intention. The wound had muscle involvement, thickened borders, and exudate leakage. This injury stands out for two aspects, firstly, its initial surface; it was a large wound (27.41 cm^2^), which reduced in size by 50% (13.88 cm^2^) at week 8 (Figure 11). Second was the favorable evolution in the percentage of tissues present in the wound bed. Table 4 shows how from week 4 and coinciding with the overcoming of the inflammatory phase of the wound, which is where the antioxidant dressing has its difference in effect with respect to other therapeutic strategies, it was possible to invert the percentage of tissue in bed, with granulation tissue predominating and devitalized tissue decreasing (Figure 12). This wound reached complete healing at week 13, outside the follow-up period established in the study.

## 4. Discussion

Wound monitoring is an essential action, providing baseline measurements, and guides us in assessing wound healing [21]. However, monitoring methods need to be accurate, reliable, and feasible in order to assess the healing process.

According to the available scientific evidence, the use of digital planimetry or digital images are highly recommended. This method provides high precision in measurements of the wound area and the tissues present in the lesion bed [3].

Based on the results of our study, the HELCOS software [22] is a complete multidimensional tool performing quantitative comparison both of the wound area and of the different types of tissues present in the wound bed throughout the follow-up period. Moreover, this information is provided through descriptive data and graphical representations. The graphs help to interpret the numerical data obtained and visually improve the interpretation of the evolution analysis performed. In addition, HELCOS [22] includes wound assessment with the validated RESVECH 2.0 scale [20]. Digital or web-based tools for wound measurement and monitoring can be a useful resource in clinical studies.

In addition, some of the data obtained in these cases align with two previously published observational studies with this antioxidant product. One was a multicenter case series developed by Castro et al. in 2017 [16] with 31 patients with acute and chronic wounds, with a follow-up period similar to ours. It describes that at the end of the follow-up period, 29% of the wounds healed completely, while in our study this was 42.85%. Regarding the variation in the RESVECH scale, Castro et al. describes a decrease in the average score of 10.16 points; similarly, in our study it was 7.89 points [16].

The other observational study mentioned was developed by Jiménez-García et al. [17], in which 31 patients with chronic wounds were included with a follow-up period of 12 weeks. The results described the evolution of wound healing evaluated by RESVECH 2.0, with a 67.8% reduction at week 12 after using the antioxidant dressing. Likewise, the percentage of wound healing increased significantly over time, and was 71% at week 12. During the follow-up time, 50% of the wounds healed completely.

One of the strengths of this study is the use of the HELCOS web-based tool, which can help clinicians differentiate between different types of tissue in the wound bed and monitor healing. This article is one of the first reports of the performance of this tool in a real context.

However, the use of this tool is not without limitations. Digital images can be affected by lighting, location, and variability when shooting, leading to an underestimation of the wound analysis [23], so it is recommended to standardize the lighting conditions for the picture.

## 5. Conclusions

The results obtained indicate that wounds monitoring helps improve healing, facilitating clinical decision-making in healthcare. For this reason, it is necessary that the measurement and monitoring methods are precise, reliable, and viable for their correct application in daily clinical practice. This is also reflected in how the use of digital applications in measuring and evaluating wounds is increasingly widespread. The HELCOS web-based system is a user-friendly and useful resource available to clinicians for wound analysis and wound healing monitoring. The antioxidant dressing used in these cases is an alternative for wound management that merits further research.

## Figures and Tables

**Figure 1 ijerph-20-04147-f001:**
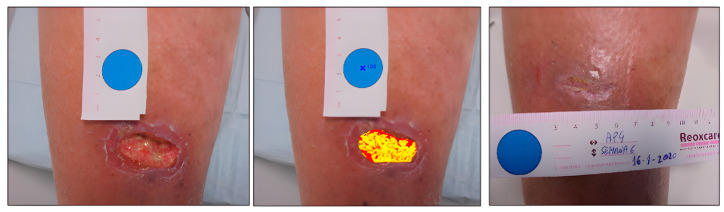
Healing evolution case 1: Baseline assessment (**left**), Baseline HELCOS analysis (**centre**), Week 6 assessment (**right**).

**Figure 2 ijerph-20-04147-f002:**
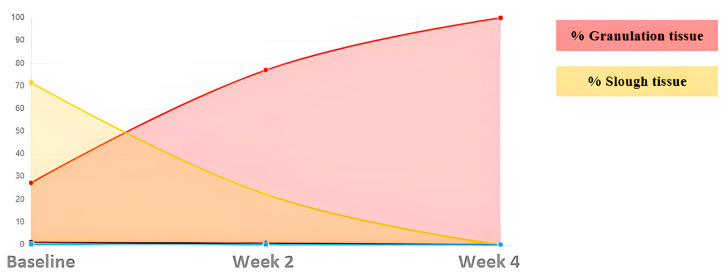
Evolution of tissue types. Case 1.

**Figure 3 ijerph-20-04147-f003:**
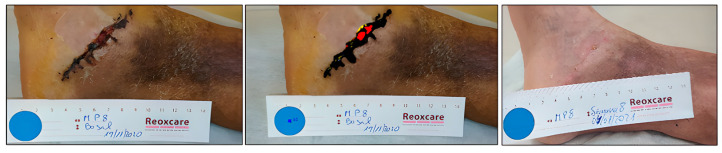
Healing evolution case 2: Baseline assessment (**left**), Baseline HELCOS analysis (**centre**), Week 8 assessment (**right**).

**Figure 4 ijerph-20-04147-f004:**
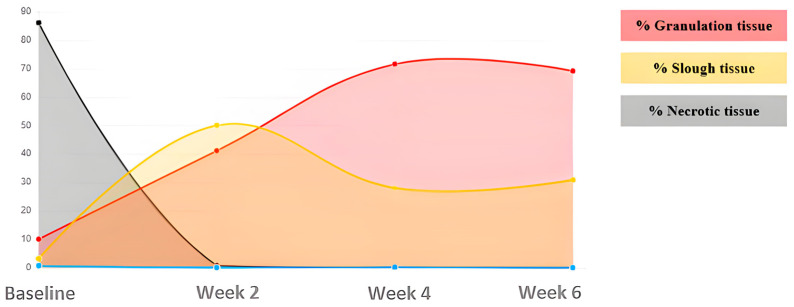
Evolution of tissue types. Case 2.

**Figure 5 ijerph-20-04147-f005:**
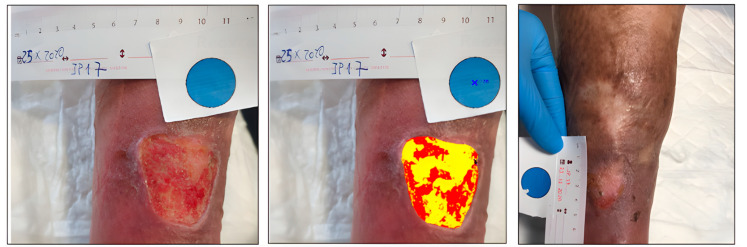
Healing evolution case 3: Baseline assessment (**left**), Baseline HELCOS analysis (**centre**), Week 8 assessment (**right**).

**Figure 6 ijerph-20-04147-f006:**
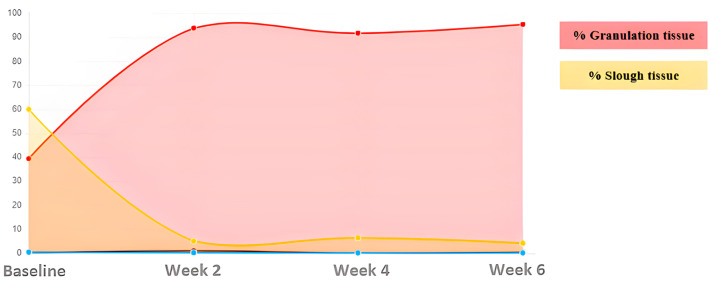
Evolution of tissue types. Case 3.

**Figure 7 ijerph-20-04147-f007:**
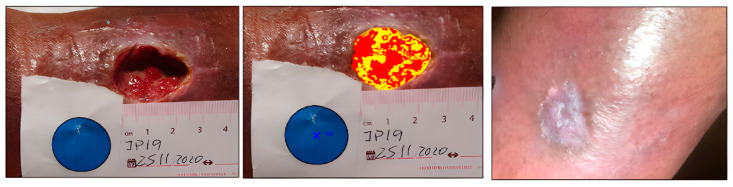
Healing evolution case 4: Baseline assessment (**left**), Baseline HELCOS analysis (**centre**), Week 4 assessment (**right**).

**Figure 8 ijerph-20-04147-f008:**
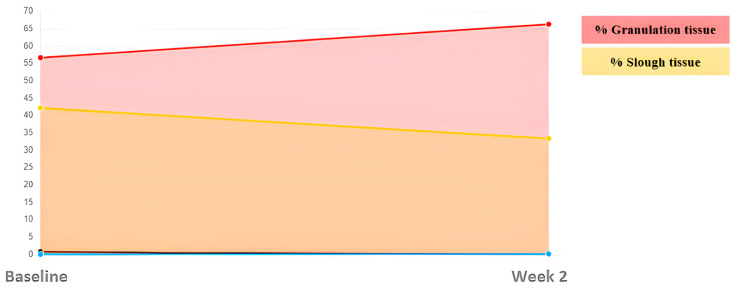
Evolution of tissue types. Case 4.

**Figure 9 ijerph-20-04147-f009:**
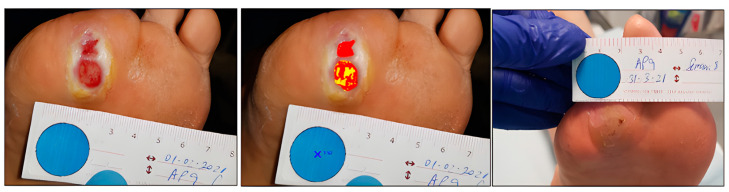
Healing evolution case 5: Baseline assessment (**left**), Baseline HELCOS analysis (**centre**), Week 8 assessment (**right**).

**Figure 10 ijerph-20-04147-f010:**
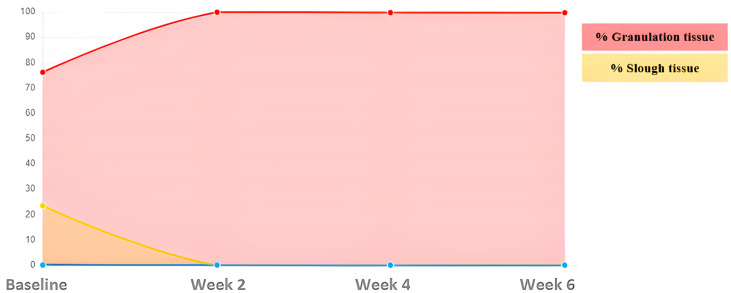
Evolution of tissue types. Case 5.

**Figure 11 ijerph-20-04147-f011:**
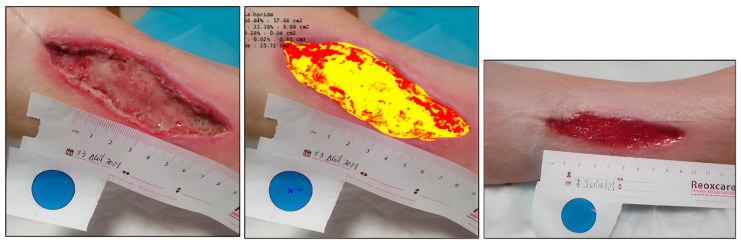
Healing evolution case 6: Baseline assessment (**left**), Baseline HELCOS analysis (**centre**), Week 13 assessment (**right**).

**Figure 12 ijerph-20-04147-f012:**
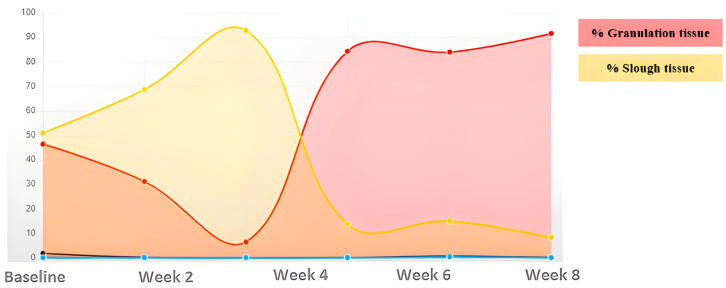
Evolution of tissue types. Case 6.

**Table 1 ijerph-20-04147-t001:** Wound characteristics.

Wound Characteristics	Mean (Standard Deviation)
Length in cm	4.75 (3.45)
Width in cm	2.86 (1.75)
Area in cm^2^	13.46 (16.88)
Area in cm^2^ **	5.64 (17.58)
Granulation tissue %	34.05 (25.46)
Devitalized tissue %	64.82 (42.69)
RESVECH 2.0 total score	15.79 (4.73)

** Median (Inter-quartile range).

**Table 2 ijerph-20-04147-t002:** Evolution in the percentage of tissue types and areas Case 1.

ASSESMENT	% Granulation	% Slough	Area (cm^2^)
Baseline	27.24	72.74	5.86
Week 2	76.99	23.01	3.74
Week 4	100	0	3.44
Week 6	27.24	72.74	5.86
Week 8	COMPLETE HEALING		

**Table 3 ijerph-20-04147-t003:** Evolution in percentage of tissue types and areas Case 2.

ASSESMENT	% Granulation	% Slough	Area (cm^2^)
Baseline	10.02	89.98	4.73
Week 2	41.1	50.73	1.49
Week 4	71.77	28.02	0.36
Week 6	69.17	30.83	0.68
Week 8	COMPLETE HEALING		

**Table 4 ijerph-20-04147-t004:** Evolution in the percentage of tissue types and areas Case 6.

ASSESMENT	% Granulation	% Slough	Area (cm^2^)
Baseline	31.1	68.9	27.41
Week 2	6.4	92.84	27.95
Week 4	84.24	13.92	21.56
Week 6	83.95	16.04	21.30
Week 8	91.57	8.46	13.88

## Data Availability

Data is contained within the article. The data presented in this study (images) are available in the figures of the article.

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
