# Peer review of "A Digital Tool for Measuring Healing of Chronic Wounds Treated with an Antioxidant Dressing: A Case Series"

_ijerph, 2023, doi:10.3390/ijerph20054147_

Round 1

Reviewer 1 Report

Manuscript ID: ijerph-2196186

Title: Measuring healing of chronic wounds treated with an antioxidant dressing

This manuscript here presents a method for measuring wound healing in patients with a antioxidant dressing using an image-based tool HELCOS. The results and hypothesis developed do not fully correlate well with experiments done to support the idea. A few experiments and analysis need to be carried out for better understanding and application before publication.

In the abstract the authors describe in the Methods:

“This article describes the potential use of this tool to improve the monitoring and follow-up of wounds and presents a case series of various chronic wounds with diverse etiology treated with an antioxidant dressing.”

Whereas in Results:

“The descriptive data showed a favorable evolution of wound healing in these wounds, indicating that antioxidant treatment is an advanced alternative in their management.”

Both the statements are contradictory with respect to what the authors are addressing in the manuscript. The authors should carefully rectify the statement to corelate each other with focus on the tool rather than the dressing.

Specific comments:

How is the data and interpretation different from earlier published results Antioxidant dressing therapy versus standard wound care in chronic wounds (the REOX study): study protocol for a randomized controlled trial - PMC (nih.gov) in terms of data collection and analysis. Was the study carried parallel to this or a separate study was carried out.

The paper starts by focusing on the wound measurement tool but deviates to focus more on the healing effect of antioxidant. The effect of dressing is important but more focus in this article should be on the tool used to measure it.

More details should be provided with respect to the development of tool and how accurately it can measure and monitor healing process.

Is this tool already available or some modifications were carried out for this study. If the tool is already available for wound healing measurement, what was the novel idea in this study.

How does the tool differentiate between granulation, necrotic tissue and slough tissue. More details should be provided so that other can replicate these results.

How was the data validated. Was there any control to compare with. If not, please include control data as well.

Methods need to be described well to allow other researchers to replicate the results obtained.

The conclusion should be modified with respect to the hypothesis and results.

Minor comments

Recent references on antioxidant material-based dressing should be updated.

Few sentences especially in figures are in non-english language and needs to be update.

Author Response

Point 1: In the abstract the authors describe in the Methods:

“This article describes the potential use of this tool to improve the monitoring and follow-up of wounds and presents a case series of various chronic wounds with diverse etiology treated with an antioxidant dressing.”

Whereas in Results:

“The descriptive data showed a favorable evolution of wound healing in these wounds, indicating that antioxidant treatment is an advanced alternative in their management.”

Both the statements are contradictory with respect to what the authors are addressing in the manuscript. The authors should carefully rectify the statement to corelate each other with focus on the tool rather than the dressing.

Response Point 1: WE have modified the section of Results in abstract:

The HELCOS software is a complete multidimensional tool performing quantitative comparison of the wound area and the different types of tissues present in the wound bed throughout the follow-up period. In the cases analised our data showed a favorable evolution of the wound healing treated with this antioxidant dressing.

Specific comments:

Point 2: How is the data and interpretation different from earlier published results Antioxidant dressing therapy versus standard wound care in chronic wounds (the REOX study): study protocol for a randomized controlled trial - PMC (nih.gov) in terms of data collection and analysis. Was the study carried parallel to this or a separate study was carried out.

Response point 2: The data published on nih.gov refer to the research protocol of the study that it was conducted as described, a comparative study. The data shown in this publication refer to a secondary analysis of data collected in the main study, with the aim was to describe the potential use of a wound measurement system. The results in the main study are in the process of publication.

Point 3: The paper starts by focusing on the wound measurement tool but deviates to focus more on the healing effect of antioxidant. The effect of dressing is important but more focus in this article should be on the tool used to measure it.

Response point 3: All information and description of the HELCOS tool is described in section 3.1 Description of software. The practical application of this tool is the quantification of the different tissues in the wound bed. This is described in section 3.2 Report, where in each clinical case the evolution with this software is shown through images over the follow-up period and in tables of evolution in percentage of tissue type.

We added a brief explication in the introduction

Point 4: More details should be provided with respect to the development of tool and how accurately it can measure and monitor healing process.

Response point 4: The details that are published about this tool are those described in the article, currently there is no further bibliography. They can be found in this bibliographic source or in the web: https://helcos.net/users/login

Verdú Soriano, J.; López Casanova, P.; Rodríguez Palma, M.; García Fernández, FP.; Pancorbo Hidalgo, PL.; Soldevilla Ágreda, JJ. HELCOS. Integrated system for wound management. Rev ROL Enferm. 2018, 41, 778-83.

Point 5: Is this tool already available or some modifications were carried out for this study. If the tool is already available for wound healing measurement, what was the novel idea in this study.

Response point 5: The tool is now available for use. The novelty of this study is that this is the first one where the tool is applied in a real context to that identify and quantify the type of tissue present in the wound bed (slough, granulation or necrotic) in the lesion bed measure the wound area. So that, to see how the wound is developing over the follow-up period. It is a tool that gives an objective, accurate and precise perspective of the evolution and not only a subjective perspective based on the vision of the clinical healthcare staff. It providing baseline measurements and guides us in assessing wound healing.

Point 6: How does the tool differentiate between granulation, necrotic tissue and slough tissue. More details should be provided so that other can replicate these results.

Response point 6: The tool differentiates between granulation, necrotic tissue and slough tissue with algorithms. The software automatically identifies tissue types by analyzing the color assigned to the type of tissues that could be present in the wound bed: red for granulation tissue, yellow for slough, and black for necrotic tissue, as can be observed you can see in the pictures.

Point 7: How was the data validated. Was there any control to compare with. If not, please include control data as well. Methods need to be described well to allow other researchers to replicate the results obtained.

Response point 7: We have revised the method section and added additional information. This article only presents data on the use of the HELCOS tool in a case series of the antioxidant dressing group. It is not a comparative study. The control group data are included in a separate article from the main study.

Point 8: The conclusion should be modified with respect to the hypothesis and results.

Response point 8: We have modified the conclusion by focusing on the results on the use of the new software to monitor wound healing.

Minor comments

Point 9: Recent references on antioxidant material-based dressing should be updated.

Response point 9: We agree that this is a research area with a lot of activity in recent years. We have searched for new studies in antioxidants and updated the references, but cannot be exhaustive

There are published results on in vitro and in vivo studies of antioxidant materials for wound healing, but there are no published clinical trials analyzing antioxidant studies on the development of wound dressings with antioxidant products, not in humans, so this dressing is novel.

Some examples of recent studies

Criollo-Mendoza, M.S.;Contreras-Angulo, L.A.;Leyva-López, N.; Gutiérrez-Grijalva,E.P.; Jiménez-Ortega, L.A.; Heredia,J.B. Wound Healing Properties ofNatural Products: Mechanisms ofAction. Molecules 2023, 28, 598.

Tyavambiza, C.; Meyer,M.; Wusu, A.D.; Madiehe, A.M.;Meyer, S. The Antioxidant and InVitro Wound Healing Activity ofCotyledon orbiculata Aqueous Extractand the Synthesized Biogenic SilverNanoparticles. Int. J. Mol. Sci. 2022,23, 16094

Verdú-Soriano J, de Cristino-Espinar M, Luna-Morales S, Dios-Guerra C, Caballero-Villarraso J, Moreno-Moreno P, Casado-Díaz A, Berenguer-Pérez M, Guler-Caamaño I, Laosa-Zafra O, Rodríguez-Mañas L, Lázaro-Martínez JL. Superiority of a Novel Multifunctional Amorphous Hydrogel Containing Olea europaea Leaf Extract (EHO-85) for the Treatment of Skin Ulcers: A Randomized, Active-Controlled Clinical Trial. J Clin Med. 2022 Feb 25;11(5):1260. doi: 10.3390/jcm11051260.

Point 10: Few sentences especially in figures are in non-english language and needs to be update.

Sorry, it was a mistake. It is already corrected

Reviewer 2 Report

This is interested research about monitoring wound healing process. 

1. From your objective is “b) report the performance of the antioxidant dressing in the treatment of hard-to-heal 86 wounds of varying etiologies.” 

However, in your experiment you don’t have a control group.  Does the innovation help wound healing faster? And in discussion part you didn’t mention about this.

2. Your want to show  that “HELCOS software” help to analysis wound status. In the figures are better show both the patient wound picture and the analysis from HELCOS software. 

And please use English in figure. 

Author Response

Point 1. From your objective is “b) report the performance of the antioxidant dressing in the treatment of hard-to-heal 86 wounds of varying etiologies.” 

However, in your experiment you don’t have a control group.  Does the innovation help wound healing faster? And in discussion part you didn’t mention about this.

Response Point 1: We have modified the objectives of this article focusing on the use of the tool for monitoring the healing.

Point 2. Your want to show  that “HELCOS software” help to analysis wound status. In the figures are better show both the patient wound picture and the analysis from HELCOS software. 

Response point 2: Yes, we agree. In each of the figures the first image corresponds to original picture of the wound and the second image to the picture analysed by the software.

Point 3: And please use English in figure. 

Response point 3: Sorry, it was a mistake. It is already corrected

Reviewer 3 Report

Dear Authors, excellent paper. Congratulations for all the hard work.

I have a few comments and recommendations that I would appreciate to have your considerations.

1. Please reconsider the title - when you read it looks like you are reading a comparative study regarding the antioxidant dressing, when your focus is on the tool and effectiveness of the dressing or a case series

2. Abstract - I would remove the numbers before the titles. Doesn't say the aim of the study/article, only speaks about the Instrument HELCOS. Method talks about the aim. Please consider in rewriting the abstract

3. Introduction is a Litle confusing because you try to answer the two aims of the study "The purpose of this article is a) to describe the potential use of a web-based wound 84 measurement system (HELCOS) for monitoring the progress of wound healing; and b) 85 report the performance of the antioxidant dressing in the treatment of hard-to-heal 86 wounds of varying etiologies. This is a secondary analysis of data collected in the main 87 study 

4. Methods - The study design is not clear, only describes recruitment - Please be more descriptive on the secondary analyses of the study. Inclusion and exclusion criteria of the first study? There is a lack of information need to be improved

5. Results withs really interesting findings, should consider that we had no information regarding the original study. need to be clarified.

6. Discussion is interesting but could be more precise on the how strengths and weaknesses of a study design should be seen in light of the kind of question the study sets out to answer

7. Conclusion could be improved and more descriptive, but is synthetic nothing to had.

Author Response

Point 1. Please reconsider the title - when you read it looks like you are reading a comparative study regarding the antioxidant dressing, when your focus is on the tool and effectiveness of the dressing or a case series

Response point 1: We have modified the title according this suggestion.

“Measuring healing of chronic wounds treated with an antioxidant dressing: a case series”.

Point 2. Abstract - I would remove the numbers before the titles. Doesn't say the aim of the study/article, only speaks about the Instrument HELCOS. Method talks about the aim. Please consider in rewriting the abstract

Response point 2: The numbers are listed in the abstract according to the journal's publication rules. The abstract has been rewritten.

Point 3. Introduction is a Litle confusing because you try to answer the two aims of the study "The purpose of this article is a) to describe the potential use of a web-based wound 84 measurement system (HELCOS) for monitoring the progress of wound healing; and b) 85 report the performance of the antioxidant dressing in the treatment of hard-to-heal 86 wounds of varying etiologies. This is a secondary analysis of data collected in the main 87 study 

Response point 3: We have modified this paragraph focusing only in one objective (the use of tool to monitor the healing).

Point 4. Methods - The study design is not clear, only describes recruitment - Please be more descriptive on the secondary analyses of the study. Inclusion and exclusion criteria of the first study? There is a lack of information need to be improved

Response point 4: The criteria of study are:

INCLUSION CRITERIA

  • Patients over 18 years of age
  • Patients with leg ulcers (venous, ischaemic, traumatic or diabetic foot ulcer)
  • Patients with dehisced surgical wounds healing by second intention
  • Patients with pressure ulcers
  • Wound area between 1 and 250 cm2

EXCLUSION CRITERIA

  • Systemic inflammatory disease or oncological disease
  • Wounds with clinical signs of infection
  • Terminal situation (life expectancy less than 6 months)
  • Ulcers from other aetiologies: tumours, infectious
  • Wounds treated with negative pressure therapy
  • Pregnancy
  • History of sensitivity or allergy to any of the components of the study dressing

Point 5. Results withs really interesting findings, should consider that we had no information regarding the original study. need to be clarified.

Response point 5: The results of the original study have not yet been published. This study show that HELCOS system allows the assessment of wound healing progression in wounds from different etiologies and at different healing phase. HELCOS data showed the adequate evolution of healing process along time in wounds treated with antioxidant dressing.

Point 6. Discussion is interesting but could be more precise on the how strengths and weaknesses of a study design should be seen in light of the kind of question the study sets out to answer

Response point 6: We had briefly explained some strengths and limitations of the study.

Point 7. Conclusion could be improved and more descriptive, but is synthetic nothing to had.

Response point 7: The conclusions have been revised and explained.
